# Pectin as a Biomaterial in Regenerative Endodontics—Assessing Biocompatibility and Antibacterial Efficacy against Common Endodontic Pathogens: An In Vitro Study

**DOI:** 10.3390/bioengineering11070653

**Published:** 2024-06-26

**Authors:** Raghda Magdy Abdelgawad, Nailê Damé-Teixeira, Katarzyna Gurzawska-Comis, Arwa Alghamdi, Abeer H. Mahran, Rania Elbackly, Thuy Do, Reem El-Gendy

**Affiliations:** 1Division of Oral Biology, Leeds School of Dentistry, St. James University Hospital, University of Leeds, Leeds LS9 7TF, UK; r.m.abdelgawad@leeds.ac.uk (R.M.A.); nailedame@unb.br (N.D.-T.); dnasaa@leeds.ac.uk (A.A.); n.t.do@leeds.ac.uk (T.D.); 2Department of Endodontics, Faculty of Dentistry, Assiut University, Assiut 83523, Egypt; 3Department of Dentistry, School of Health Sciences, University of Brasilia, Brasilia 70910-900, Brazil; 4Oral Surgery, Life Course and Medical Science, Liverpool L8 7SS, UK; k.gurzawska-comis@liverpool.ac.uk; 5Oral Biology Department, Faculty of Dentistry, King Abdulaziz University, Jeddah 21589, Saudi Arabia; 6Department of Endodontics, Faculty of Dentistry, Ain Shams University, Cairo 11566, Egypt; abeer.hasham@dent.asu.edu.eg; 7Endodontics, Conservative Dentistry Department and Tissue Engineering Laboratories, Faculty of Dentistry, Alexandria University, Alexandria 21527, Egypt; rania.elbackly@alexu.edu.eg; 8Department of Oral Pathology, Faculty of Dentistry, Suez Canal University, Ismailia 8366004, Egypt

**Keywords:** antimicrobial effect, alternative antimicrobials, biofilm, dental pulp stem cells, regenerative endodontics, pectin

## Abstract

Regenerative endodontics (REP) is a new clinical modality aiming to regenerate damaged soft and hard dental tissues, allowing for root completion in young adults’ teeth. Effective disinfection is crucial for REP success, but commonly used antimicrobials often harm the niche dental pulp stem cells (DPSCs). To our knowledge, this is the first study to explore the biocompatibility and antimicrobial potential of pectin as a potential natural intracanal medicament for REPs. Low methoxyl commercial citrus pectin (LM) (pectin CU701, Herbstreith&Fox.de) was used in all experiments. The pectin’s antibacterial activity against single species biofilms (*E. faecalis* and *F. nucleatum)* was assessed using growth curves. The pectin’s antimicrobial effect against mature dual-species biofilm was also evaluated using confocal laser scanning microscopy (CLSM) after 30 min and 7 days of treatment. The DPSC biocompatibility with 2% and 4% *w*/*v* of the pectin coatings was evaluated using live/dead staining, LDH, and WST-1 assays. Pectin showed a concentration-dependent inhibitory effect against single-species biofilms (*E. faecalis* and *F. nucleatum)* but failed to disrupt dual-species biofilm. Pectin at 2% *w*/*v* concentration proved to be biocompatible with the HDPSCs. However, 4% *w*/*v* pectin reduced both the viability and proliferation of the DPSCs. Low concentration (2% *w*/*v*) pectin was biocompatible with the DPSCs and showed an antimicrobial effect against single-species biofilms. This suggests the potential for using pectin as an injectable hydrogel for clinical applications in regenerative endodontics.

## 1. Introduction

Dental caries is the most frequently occurring oral disease and a significant cause of tooth loss worldwide, particularly among young adults. Dental decay involves the gradual demineralization of dental hard tissues and the formation of pathogenic bacterial biofilm on tooth surfaces, which might cause pulp inflammation. As the dental pulp is enclosed within the hard mineralized walls and lacks collateral circulation, inflammation typically spreads throughout the entire pulp tissue, leading to pulp necrosis [1,2]. This will allow microorganisms to invade the pulp space and establish colonies in the root canal system, causing endodontic infections. Based on the time of the microbial invasion of the pulp space, endodontic infections can be classified into three categories, namely primary, secondary, and persistent endodontic infections [3]. A primary endodontic infection is a polymicrobial infection resulting from initial microbial invasion of the pulp space and the colonization of necrotic tissues. It is mainly caused by Gram-negative anaerobic bacteria, where Fusobacterium spp. is considered one of the most frequently implicated species in this type of infection. It plays a fundamental role in endodontic flare-up cases and acts as a bridge between the primary and secondary colonizers of dental plaque [4,5]. Conversely, a persistent endodontic infection is caused by microorganisms that survive the treatment procedures and persist within the treated root canal [3]. *E. faecalis* was found to be frequently isolated in these cases and is the predominant bacteria responsible for pain and infection following endodontic therapy, with prevalence rates as high as 90% [6,7,8].

Root canal treatment is considered the gold standard treatment of choice in cases of pulp and periapical infection. This approach is based on the complete removal of the damaged pulp tissues, and both mechanical and chemical debridement of pulp space, followed by replacement with a biocompatible filling material. The primary objectives of root canal treatment are to eliminate existing infections by disinfecting the root canal system and preventing future reinfections. REP treatment can help to prevent the progression of pulp inflammation; however, it could be a challenging procedure when the pulp necrosis occurs in immature permanent teeth. An immature permanent tooth is a newly erupted tooth with an incomplete root apex, and it takes about three years for this root to develop and achieve apical closure [9]. The most frequently employed method for treating such teeth is apexification, which is a technique that uses calcium hydroxide or mineral trioxide aggregate (MTA) to induce hard tissue apical barrier formation. The main disadvantage of this approach is that immature teeth have thin dentin walls, and this treatment might increase the risk of tooth fracture. Additionally, the success rates of this treatment approach were found to range from 26% to 100% [10,11].

Alternatively, regenerative endodontic treatment (RET) is centered around repairing dental pulp tissues rather than substituting them with synthetic materials. This can result in the continuation of root development, therefore increasing its resistance to fracture. RET involves the extirpation of pulp tissue with minimum or no mechanical instrumentation, complete eradication of root canal infection, followed by the stimulation of the resident dental pulp stem cells. However, the success of this treatment mainly relies on the thorough disinfection of the root canal space using antimicrobial agents, since there is limited or no mechanical instrumentation involved [12]. To date, the most commonly used antimicrobials in endodontics fail to completely eradicate bacterial biofilms. Calcium hydroxide is considered the gold standard intracanal medication, and it has shown a high biocompatibility with dental stem cells when compared to other antimicrobials [13]. Nevertheless, it was found to have a weak action against *E. faecalis* [6]. To overcome the limitation of calcium hydroxide, antibiotic pastes were investigated as an alternative; however, their cytotoxic effect against dental stem cells was reported, which could compromise the success of the whole process [13]. Additionally, the excessive use of antibiotics in root canal treatment can worsen the situation by developing resistant microbial strains. According to the literature, up to 10% of the annual global antibiotic prescriptions are prescribed by dental care providers, which also contributes to the global antimicrobial resistance challenge [14]. Hence, finding and investigating alternative antimicrobial agents that effectively disinfect the root canal while preserving dental pulp stem cells is necessary. This approach, if successful, might also reduce the use of antibiotics in dentistry, augmenting the global antimicrobial stewardship efforts and addressing the global antimicrobial resistance challenge [11].

Another important factor in REPs is the regeneration factor and creating a cell-friendly microenvironment that can recruit and maintain niche cells for tissue regeneration. Several niche stem cells are suitable candidates to spearhead the regeneration procedure. However, the current clinical procedure employs the stem cells of apical papilla (SCAP), which play a crucial role in root development. SCAPs, located at the apex of immature roots, can be readily mobilized by inducing bleeding within the canal. However, the presence of remnants of the epithelial root sheath of Hertwig (HER) or the epithelial rests of Malassez (ERM) is imperative for SCAPs to foster the epithelial–connective tissue interaction essential for the regeneration of pulp, dentin, and cementum tissues, thereby facilitating root thickening and elongation. Nevertheless, pulp necrosis and infection in teeth with incomplete roots can compromise the viability of SCAPs or HER [14]. In light of this, the proposition of utilizing dental pulp stem cells (DPSCs) has emerged as a refined alternative, either through in situ application or via autologous transplantation following banking and ex vivo expansion.

Recently, several natural antimicrobials have emerged in the field of regenerative dentistry and acquired a high global research interest as alternatives to the widely used antibiotics and chemical agents [15]. Pectin, a natural polysaccharide complex, is found in the cell walls of higher plants. It is commonly extracted from citrus and is known to have antimicrobial properties against *Helicobacter pylori* [16]. Pectin’s antimicrobial properties have not been thoroughly evidenced yet; however, one study suggested the potential drug delivery power of pectin, when used as a film containing metronidazole, through intra-periodontal pockets to target the polymicrobial biofilm [17]. This study aimed to investigate the antimicrobial effect of pectin against the planktonic bacteria commonly isolated in both primary and persistent endodontic infections, and its effect on complex oral biofilm, in addition to its biocompatibility with human dental pulp stem cells.

## 2. Materials and Methods

### 2.1. Pectin Suspensions Preparation

Low methoxyl commercial citrus pectin powder (Classic CU 701 LM Citrus fruits DE 34–38%, Galacturonic acid content 89%, Pectin content 20–35%) was kindly provided by Herbstreith & Fox, Neuenbürg, Germany. Suspensions of different pH and concentrations were prepared to be tested for their antibacterial effect against planktonic bacterial cultures. Additionally, two different methods of sterilization were used to test their effect on the bacteriostatic potentials of pectin. In the first method, a 5% *w*/*v* suspension stock was made by dissolving the powder in distilled water followed by moist heat sterilization at 121 °C [18]. Three concentrations (4%, 2%, and 1% *w*/*v*) were then obtained by diluting the stock solution using sterile Brain–Heart Infusion broth (BHI). The pH of the original stock was measured (<5), and another solution of pH (>5) was also prepared by adding 1N of NaOH until the desired pH was attained [19]. In the second method, pectin powder was sterilized using Gamma irradiation (15 kGy–Co-60 Gamma Irradiator), and then pectin suspensions of the same concentrations were prepared by dissolving the sterile powder in sterile distilled water under sterile conditions.

### 2.2. Planktonic Bacteria Culture Conditions

In this study, *Enterococcus faecalis* (Gram-positive facultative anaerobe) and *Fusobacterium nucleatum* (ATCC10953) (Gram-negative, obligate anaerobe) were selected as they are bacterial strains commonly associated with endodontic infections [20,21]. Both strains were obtained from the Division of Oral Biology bacterial culture collection (stored in and revived from −80 °C freezer stocks). *Enterococcus faecalis* was grown in (BHI) agar anaerobically (85% nitrogen, 5% carbon dioxide, 10% hydrogen) at 37 °C for 24 h, and one loop of bacterial colonies was subcultured in 10 mL of sterile BHI broth overnight, while *Fusobacterium nucleatum* was grown in Fastidious Anaerobic Agar (FAA) anaerobically under the same conditions, and two loops of colonies were subcultured in 10 mL of BHI broth overnight.

### 2.3. Growth Assays of Planktonic Bacteria

Previously prepared pectin suspensions (4%, 2%, and 1% *w*/*v*) were inoculated with the two test strains and then incubated for 5 h (*n* = 2, *n* = 3). The optical density (OD) of 600 of each concentration was measured each hour using a spectrophotometer (Jenway™ 6305 UV/Visible Spectrophotometer, Fisher Scientific UK) to obtain a five-hour bacterial growth curve. The percentage of bacterial growth was calculated using the following equation:(time point OD-blank) − (baseline OD-blank) × 100
where it was then compared between the groups. The positive control (media with inoculum) and negative controls (sterile pectin suspensions of the tested concentrations) were included in the same conditions. Experiments were performed in biological duplicates in three independent experiments.

### 2.4. Dual-Species Biofilm Model

To investigate the antimicrobial effect of LM pectin on biofilm, a dual-species biofilm model was developed (*Enterococcus faecalis* and *Fusobacterium nucleatum)* using Calgary Biofilm Device for high-throughput antibacterial and antibiofilm screening (CBD; MBECTM Assay System, MBEC Biofilm Technology Ltd., Calgary, AB, Canada). The bacterial species were cultured, and the inoculum of each species was prepared as previously mentioned. CBD Pegs were coated with 200 µL of artificial saliva (AS) (Hog gastric mucin (Sigma, Saint Louis, MO, USA) 2.5 (g/L), NaCl (Sigma) 0.381 (g/L), KCl (Sigma) 1.114 (g/L), KH_2_PO_4_ (BDH) 0.738 (g/L), ascorbic acid (Fisher Scientific, Waltham, MA, USA) 0.002 (g/L), urea (Fisher Scientific) 9 mM, and arginine (Sigma) 5 mM) and then left in the anaerobic incubator overnight. CBD Plates were inoculated with the dual strains and then incubated anaerobically for 7 days, in which fresh BHI media were added every 24 h (*n* = 3). A solution of a low-concentration double-antibiotic paste (DAP) (1 mg/mL ciprofloxacin, 1 mg/mL metronidazole) was used in this experiment as a positive control. After 7 days, the biofilms were treated with 2% pectin solution and DAP for 30 min and for 7 days, where no changing for the media performed throughout the treatment period. Following the application of different treatments, CBD pegs were snipped off the Calgary plate with sterile pliers, washed with sterile PBS, treated with a Film-tracer Live/Dead Biofilm Viability Kit (Molecular Probes, Inc., Waltham, MA, USA), and incubated for 30 min at room temperature in the dark. The biofilm samples were scanned by confocal laser scanning microscope (Leica SP8 TCS Microsystems, Wetzlar, Germany) using a 20× water immersion objective lens. Two different areas were scanned on each biofilm sample, and 3D images for the experimental and control groups were generated. In addition, 2D images were acquired for image analysis using the maximum projection option. Biofilm viability analysis was carried out using the biofilm viability checker tool (Fiji, ImageJ 1.53t National Institutes of Health, Bethesda, MA, USA) according to Mountcastle et al., 2021. [22]

### 2.5. DPSCs Culture and Cell Expansion

Human dental pulp stem cells (DPSCs) used in this study were previously isolated from permanent wisdom teeth from three donors (Leeds Dental and Skeletal Tissue bank-DREC ethical approval no. 251121/HA/366). Cells were cultured in alpha-modified minimum essential medium (α-MEM) complete media, supplemented with 20% fetal bovine serum (FBS), 2 mM glutamine, 100 U penicillin/0.1 mg/mL streptomycin, and incubated at 37 °C, 5% CO_2_. The media was changed every 5–7 days and sub-confluent cells (70–80%) were used in this study between passages 4 and 8.

### 2.6. Pectin Preparation, Plate Coating, and Cell Seeding

Gamma irradiation-sterilized LM (pectin CU701) (Herbstreith & Fox, Germany) [23] was dissolved in sterile distilled water to obtain two concentrations of pectin viscous suspensions (2% and 4% *w*/*v*). In 24 well tissue culture plates, 200 µL pectin of 4% or 2% (*w*/*v*) was applied per well to form a coat, and the coated plates were left in the hood under UV light for 1 h, followed by overnight drying under the hood. To avoid dropping the pH of the culture media caused by pectin’s low pH, 7.5% sodium bicarbonate was added to the culture media before plating in a ratio of 1:10. DPSCs at 70–80% confluence were seeded in the previously coated plates with a seeding density of 30,000/well, and incubated at 37 °C, 5% CO2 at three different time points (24 h, 3 days, and 7 days).

### 2.7. Cytotoxicity Assays

LDH (Lactate Dehydrogenase) cytotoxicity assay (Roche Applied Science, Penzberg, Germany) was used in this study according to the manufacturer’s instructions. Culture media was collected from each well at each time point to calculate the cell death rate. The reaction mix was prepared, 50 µL of each sample was transferred to 96 well plates, and 50 µL of the reagent mix was then added to each well. The plate was incubated for 30 min at 37 °C away from light. The complete culture media without cells was used as a low control group. The high control group represented the maximum LDH release generated by adding 5 µL/well lysis solution to 100 µL of the collected sample followed by incubation for 15 min. Absorbance was read at 490 nm using a plate reader (Varioskan Flash Spectral Scanning Multimode Reader, Thermo Scientific, Waltham, MA, USA) and the percentage of cytotoxicity was calculated for each sample using the following equation:Cytotoxicity % = {(experimental absorbance value − low control absorbance value)/(high control absorbance value − low control absorbance value)} × 100

### 2.8. Cell Proliferation Assay

To measure DPSC proliferation, a WST-1 proliferation kit was used (Roche Applied Science, Penzberg, Germany) according to the manufacturer. Cells were washed with PBS, and 150 µL of plain α-MEM was added to each sample, followed by 15 µL of WST-1 reagent (1:10) and the plate was incubated for 4 h in a humidified atmosphere at 37 °C and 5% CO_2_. After incubation, 100 µL of the reaction mixture was transferred to each well of a 96 well TC plate. The absorbance was read at 450 nm using a plate reader (Varioskan Flash Spectral Scanning Multimode Reader, Thermo Scientific, USA). The complete culture media was used as the low control group, while the maximum read from 30,000 cells at the initial time point represented the high control group. The DPSC proliferation percentages were calculated according to the following equation:The proliferation of DPSCs (%) = {(experimental absorbance value − low control absorbance value)/(high control absorbance value − low control absorbance value)} × 100.

### 2.9. Cell Viability Assay

According to the manufacturer, DPSC viability was assessed using a live/dead viability/cytotoxicity kit for mammalian cells (LIVE/DEAD ^®^, Invitrogen™, Thermo Fisher Scientific, Waltham, MA, USA). Cell adherence was confirmed using a light microscope, then cells were washed twice with PBS. To create a staining solution, 4 μL ethidium stock solution and 1 μL of calcein stock solution were added to 2 mL of plain media and then vortexed to ensure thorough mixing. A volume of 200 μL of the combined live/dead assay reagents was added to each well, and the plate was then incubated for 45 min to 1 h at room temperature covered from light. After incubation, the cells were imaged using a light/fluorescent microscope using a 5× lens (Zeiss microscopy, Carl Zeiss NTS Ltd., Oberkochen, Germany).

### 2.10. Statistical Analysis

All antimicrobial experiments were carried out in triplicates and repeated 3 times (*n* = 9), whereas biocompatibility experiments were carried out in 3 biological replicates from 3 different donors.

Data were analyzed using GraphPad Prism version 9.5.1 (733). According to the data normality (Shapiro–Wilk) of each experiment, the means were compared between groups using two-way ANOVA, followed by Tukey’s multiple comparison test for data with normal distribution or Kruskal–Wallis’s test, as well as Dunn’s multiple comparisons test for the data that were not normally distributed.

## 3. Results

### 3.1. Antimicrobial Effect of Pectin

#### 3.1.1. *E. faecalis* and *F. nucleatum* Bacterial Growth Curve in Response to Different Concentrations of Pectin Suspension

Pectin sterilized using the moist heat method showed a concentration-dependent inhibitory effect against the two tested bacterial strains (*E. faecalis* and *F. nucleatum).* The growth curves showed a significant reduction in bacterial growth within the test groups according to the different concentrations in comparison to the control untreated groups (Figure 1). To confirm that the antimicrobial effect of pectin is specific and not related to its low pH, two suspensions of pectin of high and low pH (<5 and >5) were additionally tested. Pectin suspensions of concentrations 2% and 4% *w*/*v* (pH < 5) showed a significant reduction in the *E. faecalis* growth curve in comparison to both the lowest concentration group 1% *w*/*v* and the control untreated group (Figure 1a). In contrast, pectin suspensions of pH > 5 showed a different effect on the *E. faecalis* growth rate, where the 4% *w*/*v* concentration did not show any antimicrobial effect compared to the control groups, while the lower concentrations 2% and 1% *w*/*v* showed a significant increase in bacterial growth in comparison to the control group (Figure 1b).

As for the *F. nucleatum* growth curves, the pectin suspensions of pH < 5 showed a significant reduction in bacterial growth within all the tested concentration groups (4%, 2% and 1%) in comparison to the control untreated groups (Figure 1d–f). Regarding pectin suspensions of (pH > 5), only the 4% *w*/*v* concentration group showed a significant reduction in bacterial growth in comparison to both the lower concentration groups (2%, 1%) and the control untreated group (Figure 1f).

Regarding the effect of different sterilization methods on the bacteriostatic potentials of pectin, Gamma irradiation-sterilized pectin showed a remarkably significant bacteriostatic effect regardless of the concentrations in comparison to the control untreated groups (Figure 1c,f).

#### 3.1.2. Antimicrobial Effect of Pectin against Dual-Species Biofilm

In this work, the biofilms treated with pectin for both 30 min and 7 days did not show any reduction in viability (Figure 2). On the other hand, it was found that after 30 min of treatment, biofilms treated with DAP showed a significant reduction in viability in comparison to both the control and pectin-treated groups. After 7 days of treatment, both untreated and DAP-treated biofilms showed a complete death in the developed biofilm in contrast to the pectin-treated group, which was the only group that demonstrated 100% biofilm viability.

#### 3.1.3. Biocompatibility of Different Concentrations of Pectin with Dental Pulp Stem Cells (DPSCs)

DPSCs treated with the pectin coating of 4% concentration showed a significant reduction in cell viability at all the different time points compared to the untreated control group, in comparison to those cultured at 2% concentration and the untreated controls. DPSCs cultured in 2% pectin showed a significantly higher percentage of cell viability in comparison to the controls at 24 h and 7 days. Significantly higher viability was also shown within the same group at 7 days compared to 3 days (Figure 3b). The average cytotoxic effect of pectin on DPSCs was compared between the groups, and the LDH assay showed no significant difference in the LDH release showed no significant difference between the groups of cells treated with pectin at 2% and 4% concentrations compared to the untreated control cells at all the time points, with all death percentages remaining below 2% at all time points (Figure 3c).

DPSCs showed their normal spindle shape morphology in the control group and after 24 h of culture in both pectin concentrations. However, a change in cell morphology in response to both the 2% and 4% pectin coats were observed after 3 and 7 days of culture. The group of cells treated with the 2% pectin coat showed elongated cell morphology, denoting possible odontogenic differentiation, whereas cells treated with the 4% pectin showed spherical clusters, denoting a lack of attachment and the possible negative effect of higher pectin concentrations (Figure 4a).

The effect of pectin’s different concentrations on DPSC proliferation was examined using WST-1 assay at 24 h, 3 days and 7 days. There was no significant difference between the cells cultured with a 2% coat of pectin and the control group, while the 4% pectin coat group showed a significant reduction in proliferation in comparison to the untreated control group. In contrast, on day three, both concentrations of pectin coats (2% and 4%) showed a significant reduction in cell proliferation in comparison to the control group, with the 4% pectin coat group showing a higher negative effect on DPSC proliferation. On day 7, there was a statistically significant difference between the two test groups compared to the control group, with the 4% pectin coat group showing a higher negative effect on cell proliferation compared to the 2% pectin coat group (Figure 4b).

## 4. Discussion

Following the global research focus on finding alternative natural antimicrobials, this work aimed to investigate the antimicrobial potentials of LM commercial citrus pectin and its biocompatibility with dental pulp stem cells to determine its validity for future use in clinical regenerative endodontics. The antibacterial activity of pectin against various bacterial strains has long been studied in the literature since the end of the 1930s [16,18,19,24,25,26,27,28,29,30,31,32]. In 1937, Edith Hynes was the first to uncover the antibacterial potentials of pectin against *Escherichia coli* [18]. Prickett and Miller also found that the growth of *E. coli* was inhibited by 1–2% pectin in broth, depending on the pH of the medium [29]. The current study examined the antibacterial effects of LM citrus pectin with varying concentrations (4%, 2%, and 1% *w*/*v*) at two different pH levels (<5 and >5). These concentrations have been extensively studied in the literature and have demonstrated high efficacy against a variety of tested pathogens [18,19,24,29,31]. In addition, pectin is known to be extremely difficult to dissolve, hence higher concentrations require prolonged heat treatment, which may break down the pectin molecules and affect their functions [18].

Our findings demonstrated that pectin with a pH < 5 has a concentration-dependent inhibitory effect on *Enterococcus faecalis* in comparison to untreated control groups. This was aligned with previous findings that showed a bilinear relationship between the concentration of pectin and the percentage of bacterial killing [19,24]. On the other hand, when the pH > 5, pectin did not exhibit any bactericidal properties. Pectin with a low concentration showed a significant increase in bacterial growth compared to the untreated control groups. Based on prior research, it was found that the antimicrobial properties of pectin diminish entirely at pH 5 or greater. This can be attributed to pectin’s susceptibility to changes in the hydrogen ion concentration of the surrounding medium [16,18,19,26,31].

The increase in the *Enterococcus faecalis* bacterial growth in response to high pH pectin could be explained by its ability to utilize pectin as for nutrition. Numerous microorganisms can produce pectinase by utilizing pectin as a carbon source [33]. Kim and colleagues conducted research that explained the mechanism of both HM and LM pectin utilization by pure bacterial cultures of selected gut bacteria as carbon sources. Their findings indicated that the degree of esterification (DE) had a significant impact on fermentations, where carbon sources that were highly methylated resulted in lower growth rates compared to those that were less methylated [34]. Torimiro and Okonji researched the impact of pH levels on the production of pectinase by the *Bacillus* species. Their study found that the optimal pH range was between 4 and 10, and they discovered that the highest amount of pectinase was produced at pH 7 [35]. Furthermore, Larsen et al. discovered that the effect of pectin on gut microbiota varied depending on its structural features. They observed that certain bacterial taxa were impacted differently by pectin and identified several factors that may contribute to differences in microbiota composition [36].

The mechanism of the antimicrobial action of pectin has been extensively studied but not clearly understood. Several studies have postulated different theories to explain the antimicrobial potentials of pectin, e.g., Pricket and Miller suggested that H-ion concentration is the factor responsible for decreases in bacterial counts [29]. According to Thunyakipisal et al., the antibacterial effect of pectin is attributed to its composition of galacturonic acid. This acid sugar produces a carbonyl anionic charge, which can attach and form a polyelectrolyte complex with the cationic sidechain of lipopolysaccharides on the surface of Gram-negative bacterial cells. This complex interferes with and disrupts the permeability of the cell wall or membrane, ultimately inhibiting the normal function of the bacteria [37].

In agreement with the previous suggestion, in the present experiment, pectin showed a concentration-dependent inhibitory effect against the *Fusobacterium nucleatum* growth curve at the two tested pH (<5 and >5). In the groups of pectin with high pH, only the 4% *w*/*v* concentration demonstrated a bacteriostatic effect in comparison to the lower concentration groups and the control untreated group. Yousef and el-Nakeeb conducted a study that also shed light on the reason for the different responses of Gram-negative and Gram-positive bacterial strains towards pectin. The results of their study showed that all the tested Gram-negative bacteria were highly responsive to pectin’s effects, likely because they lacked pectolytic activity. The examined Gram-positive bacteria presented a more complex situation, which could be attributed to their acid fragility and the low pH (3.4) of the pectin solution employed [19,38]. Other studies related the bacteriostatic effect of pectin to the action of bioactive substances in pectin (citrus bioflavonoids, hydroxylated phenolic molecules) synthesized by plants in response to microbial infection [25]. Moreover, it has been proposed that the presence of organic compounds, such as polyphenols, flavonoids, flavonols, and phenolic acids, which are frequently found in citrus fruits as “impurities” in the final pectin extract, has a significant impact on the bacteriostatic effect of pectin [27].

Halder et al., stated that the response of bacteria to pectin could be related to pectin’s electrostatic interaction with the polysaccharides and proteins found on the surface of bacterial cells. The study found that LM pectin had a greater propensity to bind to the oppositely charged groups on bacterial surfaces due to an increased number of negatively charged carboxyl groups when compared to HM pectin. It was discovered that LM lime pectin had a significant impact on bacterial viability, possibly by affecting the Zeta potential and interfering with the pH homeostasis mechanisms like proton-pumping, alteration of cell envelope composition, and metabolic pathways [39]. These findings were confirmed by research conducted by Halder et al., who found that surface-acting agents can change the Zeta potential of *Escherichia coli* and *Staphylococcus aureus*, leading to a rise in membrane permeability and a decrease in cell viability [40].

In the context of pectin sterilization, moist heat sterilization was employed in this study, as it has been extensively used in the literature [18,19,24,29,41]. However, one study revealed that utilizing this method can notably impact the functional characteristics of pectin, particularly its capacity to form a gel. To avoid this negative impact of moist heat sterilization, the gamma irradiation sterilization method was also employed and compared in terms of its impact on the bacteriostatic effect of pectin. Irradiation was conducted at a lower dose of 15 kGy instead of the typical 25 kGy. According to Munarin et al., this dose could be employed to avoid the change in pectin physical properties caused by higher doses. However, the authors recommended that it is necessary to validate its effectiveness before implementing low-energy gamma irradiation sterilization in the healthcare and food industries [23].

Our results showed that gamma irradiation demonstrated an increase in the bacteriostatic effect of pectin, especially against *Fusobacterium nucleatum* within all the tested concentration groups in comparison to moist heat sterilization. As for *Enterococcus faecalis*, only the low-concentration groups demonstrated a notable increase in the bacteriostatic potential compared to their correspondence within the heat-sterilized pectin groups. This might be related to the previously mentioned adverse influence of heat on the functional characteristics of pectin, which consequently might have affected its antimicrobial properties. Additionally, the gamma-irradiated pectin demonstrated low pH (3.4), which was reported to play a major role in pectin’s antimicrobial mechanism of action. To our knowledge, no previous studies have compared the impact of different methods of sterilization on the antimicrobial effect of citrus pectin. However, one study reported the effectiveness of moist heat sterilization to enhance the antibacterial properties of durian-rind pectin in comparison to unsterile pectin. This was related to heat treatment, which degraded pectin and decreased pH, leading to increased antibacterial properties [41].

To understand the antimicrobial potential of pectin in a setting that mimics the clinical situation of root canal infection, its effect against complex biofilm was also evaluated. Based on the recommendation of Howard Ceri and colleagues, the Calgary Biofilm Device was employed in this work for the screening of the antimicrobial properties of pectin [42]. To our knowledge, no previous work has tested the effect of pectin on the structure of complex oral biofilms. In this experiment, confocal laser scanning microscopy (CLSM) was used to examine the antimicrobial effect of pectin (2% *w*/*v*) against dual-species biofilm. After 30 min of exposure to treatment, our results showed no significant difference between the pectin-treated and control groups in terms of bacterial cell viability. However, DAP-treated biofilms showed a significant reduction in biofilm viability. As for the 7 day treatment, both untreated and DAP-treated biofilms showed a complete death of bacteria. The complete death of bacteria within the untreated biofilms could be attributed to the deprivation of nutrients to the biofilm throughout the treatment period.

The biocompatibility of LM citrus pectin toward human dental pulp stem cells was also investigated in the current study. Based on our previous findings, pectin suspensions with the two concentrations found to show a significant bacteriostatic effect (2% and 4%) were used. Several studies have examined the biological properties of pectin and the possibility of its use in regenerative medicine due to its unique physical characteristics, biodegradability, and gelling ability [43]. According to the American Association of Endodontics (AAE), 7 days is the minimum recommended duration for inter-appointment intracanal medication application in clinical REPs [44]. Following this recommendation, the DPSCs were incubated in pectin (2% and 4% *w*/*v*)-coated plates for 24 h, 3 days, and 7 days. The cytotoxicity of the different concentrations of pectin was then determined, first, by measuring the LDH release by the DPSCs at each time point, and second, by detecting the percentage of viable cells [45].

To eliminate any possible side effects of heat treatment on pectin in terms of cytotoxicity towards DPSCs, only gamma irradiation sterilization was employed in this experiment. According to Munarin et al., LM pectin subjected to moist heat sterilization exhibited a cytotoxic effect on fibroblast cells compared to the other methods of sterilization. This was attributed to the acidification of the medium caused by heat. In contrast, when 100 mL of 0.5% (*w*/*v*) of gamma-sterilized pectin was added to the cell culture medium for 24 h and 7 days, it resulted in increased cell viability and the fibroblasts showed greater elongation and proliferation compared to the control groups [23].

Our results showed that after 24 h and 3 days of HDPSC exposure to (2% *w*/*v*) pectin coating, no cytotoxic effect was demonstrated. However, after 7 days of exposure, the cells demonstrated a slightly significant decrease in cell viability in comparison to the control group. In contrast, the HDPSC cells incubated with the 4 % pectin coat showed a spherical cluster shape with reduced viability at all time points in comparison to both the lower concentration and the control group.

Our results are in agreement with a study which reported that introducing citrus pectin samples (1, 3, and 5 mg/mL) to healthy Vero cells for 24, 48, and 72 h did not cause any cytotoxic effects but rather promoted the proliferation of treated cells compared to control untreated groups. [46] Another study also found that pectin did not have a toxic effect on A549 human epithelial cells at the doses of 0.25 mg/mL, 0.5 mg/mL, and 1 mg/mL after 24 h of exposure. Microscopic observation of cellular morphology confirmed the biocompatibility of pectin at all the concentrations tested [47].

Regarding the proliferation of DPSCs in response to 2% *w*/*v* pectin, our results showed that after 24 h, the cells showed a similar rate of proliferation compared to the control group. However, after 3 and 7 days of incubation, DPSCs demonstrated a noticeable decrease in proliferation in comparison to the control untreated cells. Upon observing the DPSCs morphology using the light and florescent microscope, it was found that cells incubated with a 2% pectin coat depicted marked morphological changes. The change in cellular morphology might explain the marked decrease in cell proliferation, as this could be an indication of odontogenic differentiation of the treated cells. According to Gurzawska et al., the favorable effect of modified potato and apple pectin nanocoatings on matrix formation, mineralization, and expression of genes (real-time PCR) related to osteoblast differentiation [48]. Another study conducted by the same team investigated how potato unmodified pectin (PU) and potato dearabinanated (PA) pectin nanocoatings affected human primary fibroblast and enhanced cell proliferation, differentiation, and extracellular matrix protein production [49].

It is worth noting that these previous studies used pectin from different sources and with different structures compared to the one used in our research. However, one study examined how citrus pectin (combined with chitosan and gelatin) affects the attachment and growth of mesenchymal stem cells. The findings showed that the CGP films were very safe for use with cells [50].

In the current experiment, after 7 days of cell incubation with pectin-coated tissue culture plates, the culture media was transformed into a gel in the coated wells. This might be caused by the addition of sodium bicarbonate to the culture media before plating. According to Yang et al., who explored the effect of pH on the LM pectin gelation mechanism, it was suggested that increasing the pH increases the gelling rate [51]. This could explain the overall decrease in DPSC viability in both test groups at the day 7 time point. In addition, the culture media within the wells were not changed at any time to be able to measure the LDH release by the cells in response to the different concentrations of pectin coatings.

While we recognize that a limitation of this study is that it was conducted entirely in vitro with limited simulation of clinical scenarios to detect the effect of pectin on dentin, we view it as a proof of concept. However, we believe that further translational investigation is warranted. Utilizing in vitro translational models with dentin discs and in vivo studies, such as the ligation model, could provide more challenging conditions pulp-periodontal involvement or the induction of periapical radiolucency, enhancing the robustness and clinical relevance of our findings.

We believe our study offers a unique contribution to the field by investigating the potential of pectin alone as an injectable hydrogel for application in RET. To the best of our knowledge, existing studies in this context predominantly utilize pectin in combination with other materials, such as GelMA [52] and chitosan [53]. In these studies, pectin was primarily added to enhance bonding.

In a study by Atila, et al., 2017 [52], a complex injectable system comprising GelMA/PecTH + PMMA/SF, which released Td and Mel, was investigated for its ability to promote cell growth and odontogenic differentiation when used in conjunction with odontogenic inductive media. Thiolated pectin was utilized in this complex system to enhance the disulfide bonds between its components, thereby improving rheological gel properties. Notably, while thiolated pectin was a key component, the study did not evaluate pectin from citrus fruits as a standalone injectable gel for regenerative endodontics.

Similarly, in another study [53], chitosan combined with hyaluronic acid or pectin was compared to a blood clot in a beagle apexification model for pulp regeneration, revascularization, and its effect on the thickening of root walls and increasing root length. Although the study demonstrated superior regeneration with blood clots compared to the composite scaffolds, the pectin–chitosan groups exhibited the highest vascularization, comparable to that of the blood clot. This observation suggests a potential contribution of pectin content to the vascularization process, albeit in comparison to other scaffold compositions.

Finally, to the best of our knowledge, this is the first study to investigate both the antimicrobial effect of esterified pectin derived from citrus fruits on bacteria typically associated with endodontic infections and its biocompatibility with dental pulp stem cells (DPSCs).

## 5. Conclusions

From this study, it could be concluded that commercial pectin from citrus fruit has proven to be biocompatible with DPSCs. In terms of its antimicrobial properties, it only showed a significant effect at low PH against planktonic bacteria; however, it failed to exert any antimicrobial action on complex endodontic-like biofilm. Given its nature as a natural polymer, pectin exhibits several advantages that make it well-suited for serving as a 3D scaffold in future tissue engineering applications. Moreover, subsequent research must focus on the integration of antimicrobials into the polymer matrix of pectin to enhance their longevity and address the limited antimicrobial impact of pectin on multispecies oral biofilms. Further work is being carried out to investigate the effect of pectin on complex biofilms in terms of the changes in bacterial taxa abundance and functional gene analysis, as well as work on the effect of pectin on gene expression and dentin formation of DPSCs within a dentin slice invitro model simulating the clinical scenario.

## Figures and Tables

**Figure 1 bioengineering-11-00653-f001:**
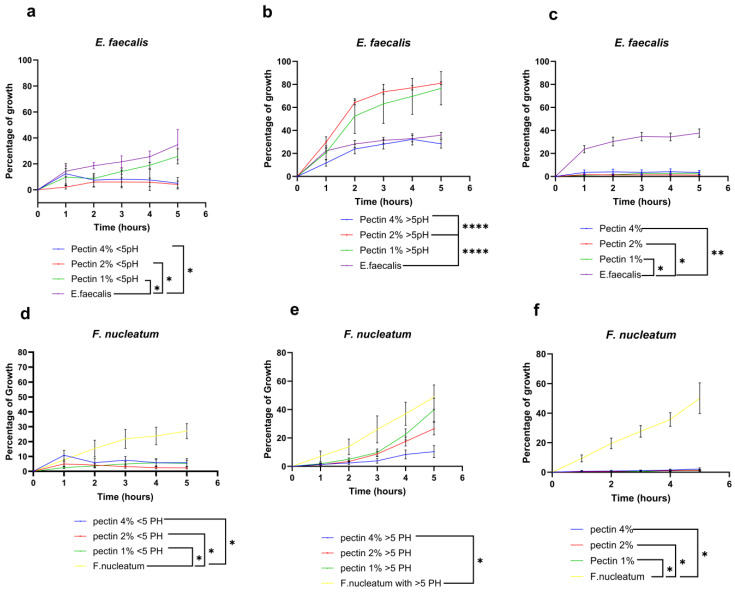
Percentage of planktonic bacterial growth. *E. faecalis and F. nucleatum* (ATCC10953) were treated with three different concentrations of pectin (4%, 2%, and 1%) (OD600). (**a**) Five hours’ growth curve of *E. faecalis* treated with low pH pectin (**b**) Five hours’ growth curve of *E. faecalis* treated with high pH pectin: (**c**) Five hours’ growth curve of *E. faecalis* treated with Gamma-sterilized pectin (**d**) Five hours’ growth curve of *F. nucleatum* treated with low pH pectin suspension. (**e**) Five hours’ growth curve of *F. nucleatum* treated with high pH pectin suspension (**f**) Five hours’ growth curve of *F. nucleatum* treated with Gamma-sterilized pectin * *p* < 0.05; ** *p* < 0.01; **** *p* < 0.0001.

**Figure 2 bioengineering-11-00653-f002:**
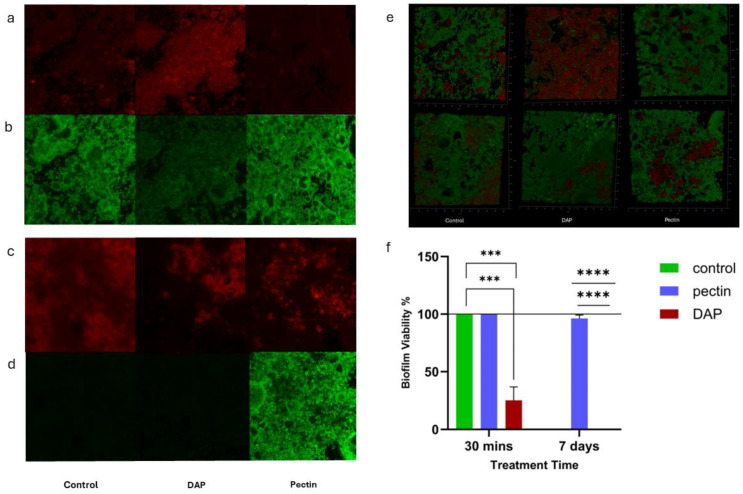
Mature biofilms after treatment with pectin 2% *w*/*v* and DAP. Live/Dead Confocal images in (**a**–**d**) 2D maximum projection, combining all 3D layers; (**e**) 3D biofilm images; (**f**) Live/Dead ratio calculated by the area of red and green within the maximum projection images. *** *p* < 0.001; **** *p* < 0.0001.

**Figure 3 bioengineering-11-00653-f003:**
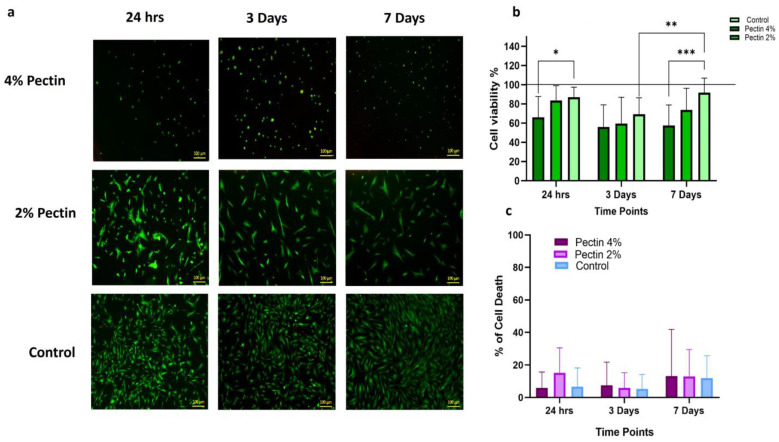
The effect of two different concentrations (4% and 2%) of pectin on the viability and percentage cell death rate of DPSCs at 24 h, 3 days and 7 days: DPSCs stained with Live/Dead staining and imaged with fluorescent microscope (**a**). Images were generated using ZEN 2012 v. 6.1.7601 software package. Calculations of percentage of live cells using ImageJ software analysis for fluorescent microscopic images (**b**). Percentage of cell death measured by LDH release from test and control groups at different time points (**c**), (* *p* = 0.0285, ** *p* = 0.0049, *** *p* = 0.0002).

**Figure 4 bioengineering-11-00653-f004:**
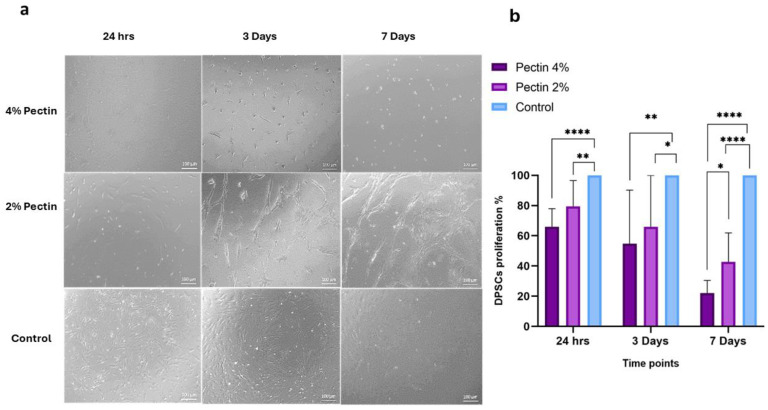
DPSC morphology and percentage of DPSC proliferation in response to different concentrations of pectin (4% and 2%) after 24 h, 3 days and 7 days in culture: light microscopic images showing changes in DPSC morphology and overall growth (**a**). The percentage of DPSC proliferation at the different time points (**b**), * *p* = 0.0285, ** *p* = 0.0049, **** *p* < 0.0001.

## Data Availability

The original contributions presented in the study are included in the article, further inquiries can be directed to the corresponding author/s.

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
