# Peer review of "Pectin as a Biomaterial in Regenerative Endodontics—Assessing Biocompatibility and Antibacterial Efficacy against Common Endodontic Pathogens: An In Vitro Study"

_bioengineering, 2024, doi:10.3390/bioengineering11070653_

Round 1
Reviewer 1 Report
Comments and Suggestions for Authors
Herein, Raghda Magdy Abdelgawad and Reem El-Gendy et al. analyzed the antibiotic effects of pectin solution to restrict the formation of biofilm caused by planktonic bacteria. They measured the influences of pectin on the growth curve of bacteria as well as on the growth of stem cells with various concentration of pectin solutions. Based on the displayed contents, the reviewer has several comments listed below.
- The title of the manuscript is too broad to conclude the key findings of the research. Authors should consider revise the title of the manuscript to present the key findings of research specifically and directly.
- The font of the titles needs to be in consistency (section 2.1). Also, the secondary sub-titles of the materials and methods parts are unnecessary.
3. Figures 1 to 3 need to be combined and better organized in a more precis way. The quality of data presentation needs to be improved. Some figures are even the screenshot, which is not acceptable.
4. Authors need to label the sample numbers n and experiment times N for each assay.
- In Figure 4, live and dead staining images need to be separately shown. Also, the data presentation is inaccurate. There seems to be missing data in Figure 4C.
- In figure 6a, little information can be concluded from the brightfield images. Meanwhile the arrangement of all figures needs to be improved.
Based on the opinions, the reviewer believes the research still need many efforts to be novel and inspiring, and the figure display and data presentation need many improvements. This manuscript may not be accepted by the journal now.
Comments on the Quality of English Languagewell written
Author Response
Reviewer 1
Herein, Raghda Magdy Abdelgawad and Reem El-Gendy et al. analyzed the antibiotic effects of pectin solution to restrict the formation of biofilm caused by planktonic bacteria. They measured the influences of pectin on the growth curve of bacteria as well as on the growth of stem cells with various concentration of pectin solutions. Based on the displayed contents, the reviewer has several comments listed below.
The title of the manuscript is too broad to conclude the key findings of the research. Authors should consider revise the title of the manuscript to present the key findings of research specifically and directly.
Response: Many thanks for the reviewer’s suggestion we have changed the title to: “Pectin as a Biomaterial in Regenerative Endodontics - Assessing Biocompatibility and Antibacterial Efficacy against Common Endodontic Pathogens: an invitro study “
The font of the titles needs to be in consistency (section 2.1). Also, the secondary sub-titles of the materials and methods parts are unnecessary.
Response: we have corrected that in the manuscript and highlighted it.
- Figures 1 to 3 need to be combined and better organized in a more precis way. The quality of data presentation needs to be improved. Some figures are even the screenshot, which is not acceptable.
Response: Figures 1 and 2 illustrate the antimicrobial effect, while Figure 3 depicts the proliferation and cell death rate of dental pulp stem cells in response to pectin coating. Therefore, we believe it is not appropriate to combine them. We attempted to merge Figures 1 and 2, but doing so compromised the image quality and resolution.
Additionally, we confirm that none of these images are screenshots.
- Authors need to label the sample numbers n and experiment times N for each assay.
Response: We have added these to the relevant sections and highlighted them.
In Figure 4, live and dead staining images need to be separately shown. Also, the data presentation is inaccurate. There seems to be missing data in Figure 4C. In figure 6a, little information can be concluded from the brightfield images. Meanwhile the arrangement of all figures needs to be improved.
Response: Thank you very much for your valuable feedback. We have taken your advice into account and have separated and rearranged the images or figures 3and 4 accordingly (pages 8 and 9). The figure legends and accompanying text have been revised to reflect these changes. We retained the brightfield images as they are important for illustrating the morphological changes.
Based on the opinions, the reviewer believes the research still need many efforts to be novel and inspiring, and the figure display and data presentation need many improvements. This manuscript may not be accepted by the journal now.
Response: We appreciate the reviewer taking the time to review our paper and provide feedback. However, we find this comment to be somewhat harsh, especially since their primary suggestions were mainly related to formatting. Regarding the novelty of our work, to the best of our knowledge, the exploration of the biocompatibility and antimicrobial effects of this specific type of pectin derived from citrus fruit, using relevant cells, bacteria, and biofilm for potential clinical applications in regenerative endodontic procedures (REP), has not been previously reported in the literature.

Reviewer 2 Report
Comments and Suggestions for Authors
Dear authors,
I evaluated the article titled “Pectin as a Biomaterial for Regenerative Endodontics: Biocompatibility and Antibacterial Activity”. Its goal was “explores the biocompatibility and antimicrobial potential of pectin, as a 21 natural potential intracanal medicament for REPs.”
- I recommend the authors include in the title that it was an in vitro study.
I read the article and I extremely appreciate the level of the study. It has a good and valid subject, which is interesting for the evolution of treatments.
The abstract is well-presented.
The introduction and methodology were well-written and clear.
No questions about the results, discussion and conclusion (very well described).
Author Response
Reviewer 2
I evaluated the article titled “Pectin as a Biomaterial for Regenerative Endodontics: Biocompatibility and Antibacterial Activity”. Its goal was “explores the biocompatibility and antimicrobial potential of pectin, as a 21 natural potential intracanal medicament for REPs.”
- I recommend the authors include in the title that it was an in vitro study.
Response: Many thanks for the reviewer’s suggestion we have changed the title to: “Pectin as a Biomaterial in Regenerative Endodontics - Assessing Biocompatibility and Antibacterial Efficacy against Common Endodontic Pathogens: an invitro study “
I read the article and I extremely appreciate the level of the study. It has a good and valid subject, which is interesting for the evolution of treatments.
The abstract is well-presented.
The introduction and methodology were well-written and clear.
No questions about the results, discussion and conclusion (very well described).
Response: Thank you very much for your feedback. We greatly appreciate your encouraging comments.

Reviewer 3 Report
Comments and Suggestions for Authors
This current study aims to evaluate the antimicrobial effect of Pectin against planktonic bacteria commonly isolated in both primary and persistent endodontic infections, its effect on complex oral biofilm, and its biocompatibility with human dental pulp stem cells.
This research is under the scope of this journal; the topic is relevant for readers, and this research deals with potentially significant knowledge to the field. And It will be important of knowledge. The topic is relevant for readers and this study deals with potentially significant knowledge to the field and opens new way for future studies.
However, some aspects need to be improved in the manuscript:
(Keywords)
* Please order the keywords / Mesh terms alphabetically
(Introduction)
What is the importance of this review study? What is the gap in this field of literature?
* You do not think this study is included in the others already done? Which results are comparable with other studies? What has this study been new?
For Regenerative endodontic procedures (REPs), the underlying strategies to promote the growth of new tissues in pulp canal space are based on 4 fundamental assumptions (1. Effective endodontic disinfection/antisepsis 2. Recruitment of undifferentiated mesenchymal stem cells (MSCs); 3. Creation of a scaffold that allows the growth of new tissue 4. Appropriate coronal sealing to prevent reinfection). A study in JOE was studying the Lyophilized hydrogel Pectin scaffold in the dentistry application in pulp regeneration, it was investigated in an animal study of the usage of scaffolds with biocompatibility for dental pulp regeneration (Regenerative Dentistry). HA was added inside the root canal dentine walls to see the recovery of dental tissues was the vascularisation. But after, 13 weeks the scaffolds were maintained inside of the canal. In some cases the lyophilized hydrogel scaffolds was obstacle for ingrowth the apical tissues to the pulp space.
Meschi in IEJ (2023) highlighted the insufficient evidence to firmly support regeneration/revitalization for apical periodontitis in (im)mature permanent teeth. Please read, https://doi.org/10.1111/iej.13778. Discuss this point with your results AND also in the introduction.
There are doubts about the survival of SCAPS. The dental papilla, when the root is formed, is in the apical zone, called the papilla apical (stem cells from apical papilla - SCAPs) that remain until the apical closure of the root. Also, these, cells can the survival of SCAPs at the infection of the tooth, (in REPs animal study and a Clinical Case HERS and the apical papilla are two embryologic structures that coordinate all the radicular development through epithelial–mesenchymal interactions. The apical papilla was a reservoir for mesenchymal stem cells (SCAPs) fundamental for root development of immature teeth. Normal dentin–pulp complex development requires the survival of HERS, ERM, and SCAP. So it has been seen as essential to describe also about REPs.
(Materials & Methods)
* When mentioning materials or devices: for some of them, you don't mention the manufacturer at all, for some you mention only the manufacturer, for some the manufacturer and city, for some you mention the manufacturer and city/ country.
* How was the sample calculated? Did the authors perform a power analysis to evaluate if this sample size was appropriate?
* How many operators performed the experiments? How many times was the experimental in vitro repeated?
* Improve the resolution quality of all figures and graphs (and a presentation). The font/language in the figure/caption is different from the text. Please, standardize the size and the font in the figures and charts with the font of the manuscript.
(Discussion)
* Please, identified what was the strength(s) and limitations of this study? And also, implications for future perspectives. About the effect on the dentin.
Author Response
Reviewer 3
This current study aims to evaluate the antimicrobial effect of Pectin against planktonic bacteria commonly isolated in both primary and persistent endodontic infections, its effect on complex oral biofilm, and its biocompatibility with human dental pulp stem cells.
This research is under the scope of this journal; the topic is relevant for readers, and this research deals with potentially significant knowledge to the field. And It will be important of knowledge. The topic is relevant for readers and this study deals with potentially significant knowledge to the field and opens new way for future studies.
However, some aspects need to be improved in the manuscript:
(Keywords)
* Please order the keywords / Mesh terms alphabetically
Response: many thanks for the reviewer’s feedback, this has been corrected as suggested.
(Introduction)
What is the importance of this review study? What is the gap in this field of literature?
Response:
We thank the reviewer for their comment, and we would like to highlight the research gap that we have already addressed in the following paragraph in page 2:
“To date, the most commonly used antimicrobials in endodontics fail to completely eradicate the bacterial biofilms. Calcium hydroxide is considered the gold standard intracanal medication, it showed a high biocompatibility with dental stem cells when compared to other antimicrobials (13) Nevertheless, it was found to have a weak action against E. faecalis, (7). To overcome the limitation of calcium hydroxide, antibiotic pastes were investigated as an alternative however, their cytotoxic effect against dental stem cells was reported, which could compromise the success of the whole process. (13). Additionally, the excessive use of antibiotics in root canal treatment can worsen the situation by developing resistant microbial strains. According to the literature up to10% of the annual global antibiotic prescriptions are prescribed by dental care providers, which will also contribute to the global antimicrobial resistance challenge (14). Hence, finding and investigating alternative antimicrobial agents that effectively disinfect the root canal while preserving dental pulp stem cells is necessary. This approach if successful might also reduce the use of antibiotics in dentistry augmenting the global antimicrobial stewardship efforts and addressing the global antimicrobial resistance challenge.
Recently, several natural antimicrobials have been emerging in the field of regenerative dentistry and acquired a high global research interest as alternatives to the widely used antibiotics and chemical agents(15). Pectin, a natural polysaccharide complex, is found in the cell walls of higher plants. It is commonly extracted from citrus and is known to have antimicrobial properties against Helicobacter pylori (16). Pectin’s antimicrobial properties were not thoroughly evidenced yet; however, one study suggested the potential drug delivery power of pectin when used as a film containing metronidazole through intra-periodontal pockets to target the polymicrobial biofilm (17).”
* You do not think this study is included in the others already done? Which results are comparable with other studies? What has this study been new?
We believe our study offers a unique contribution to the field by investigating the potential of pectin alone as an injectable hydrogel for application in RET. To the best of our knowledge, existing studies in this context predominantly utilize pectin in combination with other materials, such as GelMA (DOI: 10.1016/j.colsurfb.2022.113078) and chitosan (DOI: 10.1016/j.joen.2017.03.005). In these studies, pectin was primarily added to enhance bonding.
To further elaborate on this distinction, we have incorporated additional paragraphs and references into the discussion section of our manuscript page :
……
In a study by (DOI: 10.1016/j.colsurfb.2022.113078) , a complex injectable system comprising GelMA/PecTH+PMMA/SF, releasing Td and Mel, was investigated for its ability to promote cell growth and odontogenic differentiation when used in conjunction with odontogenic inductive media. Thiolated pectin was utilized in this complex system to enhance the disulfide bonds between its components, thereby improving rheological gel properties. Notably, while thiolated pectin was a key component, the study did not evaluate pectin from citrus fruits as a standalone injectable gel for regenerative endodontics.
Similarly, in another study (DOI: 10.1016/j.joen.2017.03.005), chitosan combined with hyaluronic acid or pectin was compared to a blood clot in a beagle apexification model for pulp regeneration, revascularization, and its effect on the thickening of root walls and increasing root length. Although the study demonstrated superior regeneration with blood clots compared to the composite scaffolds, the pectin-chitosan groups exhibited the highest vascularization, comparable to that of the blood clot. This observation suggests a potential contribution of pectin content to the vascularization process, albeit in comparison to other scaffold compositions.
Finally, to the best of our knowledge, this is the first study to investigate both the antimicrobial effect of esterified pectin derived from citrus fruits on bacteria typically associated with endodontic infections and its biocompatibility with dental pulp stem cells (DPSCs).
For Regenerative endodontic procedures (REPs), the underlying strategies to promote the growth of new tissues in pulp canal space are based on 4 fundamental assumptions (1. Effective endodontic disinfection/antisepsis 2. Recruitment of undifferentiated mesenchymal stem cells (MSCs); 3. Creation of a scaffold that allows the growth of new tissue 4. Appropriate coronal sealing to prevent reinfection). A study in JOE was studying the Lyophilized hydrogel Pectin scaffold in the dentistry application in pulp regeneration, it was investigated in an animal study of the usage of scaffolds with biocompatibility for dental pulp regeneration (Regenerative Dentistry). HA was added inside the root canal dentine walls to see the recovery of dental tissues was the vascularisation. But after, 13 weeks the scaffolds were maintained inside of the canal. In some cases the lyophilized hydrogel scaffolds was obstacle for ingrowth the apical tissues to the pulp space.
Response: we thank the reviewer for highlighting this study. We have made an effort to locate and include it in our manuscript. However, we were unable to find the specific study based solely on the provided description. To ensure accurate inclusion, could the reviewer kindly provide additional details, such as the DOI number or full title and publication year? This information would greatly assist us in incorporating the study into our work.
Meschi in IEJ (2023) highlighted the insufficient evidence to firmly support regeneration/revitalization for apical periodontitis in (im)mature permanent teeth. Please read, https://doi.org/10.1111/iej.13778. Discuss this point with your results AND also in the introduction.
Response: we believe that the suggested study is very valuable for clinical assessment of REPs as a clinical modality in the field, so we have included it in our introduction citation page..:
However, we were unable to include it in our discussion as it is beyond the scope of the current study
There are doubts about the survival of SCAPS. The dental papilla, when the root is formed, is in the apical zone, called the papilla apical (stem cells from apical papilla - SCAPs) that remain until the apical closure of the root. Also, these, cells can the survival of SCAPs at the infection of the tooth, (in REPs animal study and a Clinical Case HERS and the apical papilla are two embryologic structures that coordinate all the radicular development through epithelial–mesenchymal interactions. The apical papilla was a reservoir for mesenchymal stem cells (SCAPs) fundamental for root development of immature teeth. Normal dentin–pulp complex development requires the survival of HERS, ERM, and SCAP. So it has been seen as essential to describe also about REPs.
Response: we appreciate this excellent insight from the reviewer and we believe that touching on the role and niche of the different stem cells is important to justify our choice of dental pulp stem cells for this study, hence we have added the following paragraph and cited this reference https://doi.org/10.3390/app9193942 in our introduction pages 2 and 3 :
Another important factor in REPs is the regeneration factor and creating a cell friendly microenvironment that can recruit and maintain niche cells for tissue regeneration. Several niche stem cells are suitable candidates to spearhead the regeneration procedure. However, what the current clinical procedure employs the stem cells of apical papilla (SCAP)which play a crucial role in root development. SCAPs, located at the apex of immature roots, can be readily mobilized by inducing bleeding within the canal. However, the presence of remnants of the epithelial root sheath of Hertwig (HER) or the epithelial rests of Malassez (ERM) is imperative for SCAPs to foster the epithelial-connective tissue interaction essential for the regeneration of pulp, dentin, and cementum tissues, thereby facilitating root thickening and elongation. Nevertheless, pulp necrosis and infection in teeth with incomplete roots can compromise the viability of SCAPs or HER https://doi.org/10.3390/app9193942. In light of this, the proposition of utilizing dental pulp stem cells (DPSCs) emerges as a refined alternative, either through in situ application or via autologous transplantation following banking and ex vivo expansion.
(Materials & Methods)
* When mentioning materials or devices: for some of them, you don't mention the manufacturer at all, for some you mention only the manufacturer, for some the manufacturer and city, for some you mention the manufacturer and city/ country.
Response: this has been corrected and highlighted in the resubmitted manuscript
* How was the sample calculated? Did the authors perform a power analysis to evaluate if this sample size was appropriate?
Response: we have based our sample size on previous studies that were accepted for publications. We have repeated experiments 3 times using 3 replicates (total of 9 samples / group) for microbiological studies. We have also used 3 replicates and repeated viability experiments using DPSCs form 3 different donors (total of 9 samples/ group). This sample size has been recognised as sufficient in many publications with biomaterials characterisation.
* How many operators performed the experiments? How many times was the experimental in vitro repeated?
Response: this is a PhD study, so only the PhD student carried out the lab experiments.
We have repeated experiments 3 times using 3 replicates (total of 9 samples / group) for microbiological studies. We have also used 3 replicates and repeated viability experiments using DPSCs form 3 different donors (total of 9 samples/ group).
* Improve the resolution quality of all figures and graphs (and a presentation). The font/language in the figure/caption is different from the text. Please, standardize the size and the font in the figures and charts with the font of the manuscript.
Response: many thanks for the reviewers comments we have rearranged the figures to have more clarity. Re the figure captions, we have used the journal template, so font sizes and formats were dictated by the journal template. I am sure all these font and formatting issues will be sorted out in the proofs, if this manuscript is accepted for publication.
(Discussion)
* Please, identified what was the strength(s) and limitations of this study? And also, implications for future perspectives. About the effect on the dentin.
Response: we have added the following statement to the discussion page 14 to cover the limitation of the study
While we recognize that the limitation of this study was that it was conducted entirely in vitro with limited simulation of clinical scenarios to detect the effect of pectin on dentin, we view it as a proof of concept. However, we believe that further translational investigation is warranted. Utilizing in vitro translational models with dentin discs and in vivo studies, such as the ligation model, could provide more challenging conditions pulp-periodontal involvement or the induction of periapical radiolucency, enhancing the robustness and clinical relevance of our findings.
This statement covers the strength of the study:
Finally, to the best of our knowledge, this is the first study to investigate both the antimicrobial effect of esterified pectin derived from citrus fruits on bacteria typically associated with endodontic infections and its biocompatibility with dental pulp stem cells (DPSCs).
This statement was added to the conclusion: Further work is being done to investigate the effect of pectin on complex biofilms in terms of the changes in bacterial taxa abundance and functional gene analysis. As well as work on the effect of pectin on gene expression and dentin formation of DPSCs within a dentin slice in vitro model simulating the clinical scenario.

Round 2
Reviewer 3 Report
Comments and Suggestions for Authors
The authors improve the article. Congratulations!